# Cathelicidin: Insights into Its Impact on Metabolic Syndrome and Chronic Inflammation

**DOI:** 10.3390/metabo14120672

**Published:** 2024-12-02

**Authors:** Alina Delia Popa, Andreea Gherasim, Lavinia Caba, Otilia Niță, Mariana Graur, Laura Mihalache, Lidia Iuliana Arhire

**Affiliations:** 1Internal Medicine II Department, Faculty of Medicine, University of Medicine, and Pharmacy “Grigore T. Popa”, 700115 Iasi, Romania; alina.popa@umfiasi.ro (A.D.P.); otilia.nita@umfiasi.ro (O.N.); laura.mihalache@umfiasi.ro (L.M.); lidia.graur@umfiasi.ro (L.I.A.); 2Department of Medical Genetics, Faculty of Medicine, University of Medicine and Pharmacy “Grigore T. Popa”, 16 University Street, 700115 Iasi, Romania; lavinia.caba@umfiasi.ro; 3Faculty of Medicine and Biological Sciences, University “Ștefan cel Mare” of Suceava, 720229 Suceava, Romania; graur.mariana@gmail.com

**Keywords:** metabolic syndrome, insulin resistance, cathelicidin, LL-37

## Abstract

**Background/Objectives**: LL-37 is associated with metabolic syndrome (MetS), a constellation of risk factors comprising obesity, insulin resistance (IR), dyslipidemia, and hypertension, which elevates the risk of cardiovascular disease and type 2 diabetes. **Methods**: In this narrative review, we analyzed the literature focusing on recent developments in the relationship between cathelicidin and various components of MetS to provide a comprehensive overview. **Results**: Studies have shown that LL-37 is linked to inflammation in adipose tissue (AT) and the development of IR in obesity. Cathelicidin can enhance inflammation by activating pro-inflammatory genes, as well as modulate the inflammatory response. The mechanisms of IR include the activation of complex signaling pathways that induce inflammation and reduce insulin signaling in adipocytes. The activation of Toll-like receptors (TLRs) by cathelicidin stimulates the secretion of pro-inflammatory cytokines, contributing to the disruption of insulin function in adipose cells. Cathelicidin also influences lipid metabolism, with recent research showing a negative relationship between LL-37 levels and HDL cholesterol. Therefore, LL-37 is involved not only in the regulation of inflammation but also in lipid metabolism, potentially aggravating the cardiovascular complications associated with MetS. **Conclusions**: Cathelicidin plays a crucial role in regulating the balance between inflammatory and anti-inflammatory responses in MetS. Understanding the impact of LL-37 on these mechanisms may unveil novel approaches for addressing MetS and its associated complications.

## 1. Introduction

A few decades ago, Reaven proposed that insulin resistance (IR) plays a role not only in type 2 diabetes mellitus (T2DM) but also in cardiovascular disease (CVD) [1]. He described a cluster of associated abnormalities, later termed “metabolic syndrome X,” including IR, hyperinsulinemia, dysglycemia, dyslipidemia, and hypertension. Metabolic syndrome (MetS) is not a disease itself but rather a collection of risk factors that increase the likelihood of developing CVD and T2DM. The diagnosis is based on five indices: waist circumference, fasting blood glucose, triglyceride levels, HDL cholesterol, and blood pressure. In 2009, Alberti et al. defined MetS globally, requiring three abnormal findings, such as elevated blood pressure, dyslipidemia, elevated blood glucose, and central obesity, to diagnose the condition [2].

The pathophysiology of MetS includes many inherited and acquired factors associated with IR and chronic low-grade inflammation [3]. Metabolic disorders, including obesity, T2DM, dyslipidemia, and CVD, significantly affect health [4]. If untreated, MetS is strongly linked to a heightened risk of developing diabetes and CVD. However, numerous facets of this therapeutic entity remain poorly understood [3].

The pathophysiology of MetS involves many pathways that are only partially elucidated. An ongoing debate concerns whether the components of MetS are discrete diseases or arise from a common underlying process. Genetic, epigenetic, behavioral, and environmental variables significantly contribute to the onset of MetS. Excessive caloric intake, resulting in visceral fat accumulation, is a critical catalyst [3]. IR, chronic inflammation, and neurohormonal activation are essential processes that promote the development of MetS and its progression to CVD and T2DM. A fundamental component of these disorders is chronic low-grade inflammation, characterized by histological alterations in tissue and phenotypic changes in immune cells [4]. Chronic low-grade inflammation and immune system dysfunction present in MetS contribute to worsening prognosis, amplifying adverse effects, and increasing cardiovascular risk [5]. A major challenge is identifying systemic biomarkers that indicate local inflammation linked to metabolic dysfunction, which could enable the early detection of subclinical changes and the prevention of systemic disease development [4].

The cathelicidin gene, cathelicidin antimicrobial peptide (CAMP), is expressed in immune cells such as neutrophils (where it was first identified by Wang [6]), monocytes, macrophages, dendritic cells, natural killer (NK) and lymphocytes cells [7,8,9], as well as in epithelial cells of the skin, respiratory system, and gastrointestinal tract [7,9,10]. Gene expression is inhibited by interferon γ (IFN-γ), interleukin 6 (IL-6), bacterial exotoxins, glucocorticoids, and activated transendothelial migration. Conversely, expression is stimulated by IL-17A, tumor necrosis factor α (TNF-α), Toll-like receptor (TLR) agonists, insulin-like growth factor-1 (IGF-1), 1,25-dihydroxy vitamin D, endoplasmic reticulum (ER) injury and stress [7,10]. Cathelicidin, known as LL-37 or human cationic antimicrobial protein 18 kDa (hCAP18) in humans and Camp (cathelin-related antimicrobial peptide) in mice, has been studied for its role in innate immunity and antimicrobial defense. Due to recent progress in understanding the pathways involved in low-grade inflammation and CAMP expression in AT, studies have focused on the role of LL-37 in developing IR associated with MetS. LL-37 was proposed as a marker of MetS [11], and growing evidence supports the association between cathelicidin and the components of MetS. However, little is known about its specific role in regulating key pathways linked to metaflammation (chronic inflammation due to obesity and MetS) and IR. This narrative review aims to synthesize and critically evaluate the current evidence on the role of cathelicidin in MetS, emphasizing key findings, identifying literature gaps, and suggesting future research directions.

## 2. Overview of Metabolic Syndrome

### 2.1. Inflammation and AT

AT is located subcutaneously, where it stores approximately 80% of total body fat, while visceral AT (VAT) accounts for 10–20% and is located around internal organs [12]. AT is composed of mature adipocytes, AT-derived progenitor cells (APCs), and a stromal vascular fraction (SVF), which is considered the “local immune system” and contains endothelial and immune cells such as macrophages, CD4+ Th2 cells, and eosinophils [12,13]. Although mature adipose cells represent only 20–40% of the resident cells in AT, they occupy almost the entire tissue, with their size playing an important role in the relationships between cellular functions and pathological conditions [12]. AT remodeling in obesity involves an increase in the size and number of adipocytes, infiltration by immune cells, hypoxia, and mitochondrial and ER dysfunction [14,15]. AT dysfunction contributes to metabolic disease through inflammation, increased free fatty acids (FFAs), and altered adipokine secretion [16].

SAT has a large lipid storage capacity, and the recruitment of new APCs prevents hypertrophic expansion and protects against metabolic diseases. When SAT reaches its expansion limit, hypertrophic obesity, inflammation, and IR occur [12]. Studies have shown that visceral adipocyte size is more strongly associated with IR than SAT size [17]. Visceral obesity presents a greater metabolic risk compared to subcutaneous obesity, which may even have a protective effect [12].

Increased fat cell size in SAT has been associated with IR, independent of body mass index (BMI), in non-diabetic individuals [18]. Compared to insulin-sensitive individuals with a similar BMI, individuals with IR are characterized by an increase in the size of large mature adipocytes, a higher proportion of small adipocytes, and a decrease in the expression of differentiation markers such as peroxisome proliferator-activated receptor gamma 2 (PPARγ 2), glucose transporter type 4 (GLUT4), and adiponectin, suggesting impaired adipogenesis [12,19]. Recent studies found reduced SAT in individuals possessing risk genes for T2DM and IR [19,20]. Bone morphogenetic protein 4 (BMP4) is important for the commitment of mesenchymal progenitor cells to the adipogenic lineage, while Gremlin-1 (GREM1)—an important antagonist of BMP4—is increased in hypertrophic obesity, suggesting that impaired progenitor cell commitment may be a cause of reduced adipogenesis [21]. In hypertrophic obesity, adipocytes develop a proinflammatory profile, and TNFα inhibits the adipogenic potential of adipose progenitor cells and induces a macrophage-like phenotype [12,22].

### 2.2. Insulin Resistance

Multiple studies have established a strong correlation between obesity, insulin resistance, and chronic low-grade inflammation. The first inflammatory cells attracted to AT are neutrophils, followed by monocytes/macrophages, and then by B, T, and NK cells. Macrophage populations in AT can dictate pro- or anti-inflammatory conditions, with M1 macrophages releasing pro-inflammatory cytokines such as TNF-α, IL-1β, IL-12, and IL-23, while M2 macrophages produce anti-inflammatories cytokines such as IL-10 [13]. Weight gain promotes the recruitment and aggregation of a population of CCR2+ macrophages, which highly express genes associated with the M1 phenotype and accumulate near necrotic adipocytes. These macrophages are characterized by high levels of the inflammatory markers integrin-alpha-X (CD11c) and TLR4 [13]. The secretion of pro-inflammatory factors, such as TNFα and IL-6, affects AT function and produces systemic effects, promoting IR by activating stress kinases. TNF-α activates the MAPK/c-Jun N-terminal kinase (JNK) pathway, which phosphorylates insulin receptor substrate 1 (IRS-1), blocking insulin signaling in muscle cells [16]. JNK1 and JNK3 phosphorylate insulin receptor substrates IRS1 and IRS2, disrupting insulin signaling in AT [23]. Cytokines and growth factors activate the JAK/STAT (Janus kinase signal transducer and activator of transcription) signaling pathway which interacts with the MAPK and PI3K (phosphoinositide 3-kinase) pathways, thereby affecting IR [23]. TNF-α influences numerous processes such as lipogenesis, lipolysis, adipogenesis, mitochondrial function, and ER stress [16]. ER stress contributes to metabolic dysfunction and IR through mechanisms involving the activation of nuclear factor kappa-light-chain-enhancer of activated B cells (NF-κB), which promotes inflammation and fat accumulation [23]. Activation of the NLRP3 inflammasome by metabolic stress signals such as ceramides, excess ER Ca^2+^, and mitochondrial oxidative stress leads to increased release of IL-1β, which impairs insulin signaling in adipocytes, decreasing glucose uptake and lipogenesis, while increasing lipolysis (Figure 1). High-fat diets are associated with inflammasome activation and IL-1β production, and genetic knockout of NLRP3 in mice protects against obesity-induced IR and hepatic steatosis [16].

Obesity and IR are characterized by high rates of basal lipolysis, which increases circulating levels of FFAs. These FFAs can induce oxidative stress and mitochondrial dysfunction, contributing to lipid accumulation in organs such as muscle and liver—a process known as lipotoxicity, which inhibits insulin signaling [16].

FFAs stimulate serine kinases in hypertrophic and dysfunctional AT, inhibiting insulin receptor signaling pathways and amplifying IR. FFAs combine with TLR4 in proinflammatory M1 macrophages, triggering an excessive release of proinflammatory adipokines such as leptin, resistin, and TNF-α [24]. In the setting of IR, enhanced lipolysis from AT leads to an increased release of FFAs into the circulation, which reach the liver via the portal vein [25]. The liver uses FFAs for the de novo synthesis of triglycerides and the production of very low-density lipoprotein (VLDL). However, when the liver surpasses its capacity to oxidize fatty acids and secrete VLDL, hepatocytes accumulate triglycerides, resulting in hepatic steatosis. Increased concentrations of FFAs in the blood lead to the overproduction of VLDL, inhibiting the clearance of these particles, resulting in hypertriglyceridemia and an increase in small and dense LDL particles, which play an increased atherogenic role [26]. These processes promote hyperglycemia and hyperinsulinemia, which are critical elements in the development of pathological dyslipidemia [24].

## 3. LL-37 and Obesity

The main source of cathelicidin in humans is neutrophils [27], where it is stored as the precursor hCAP18 and subsequently cleaved extracellularly to form LL-37 during systemic immune responses. AT contributes to cathelicidin production as a secondary or induced source, especially during infection or inflammation [27]. Within AT, preadipocytes appear to be the main cell type responsible for cathelicidin synthesis [28].

In 2015, Zhang et al. [28] were the first to demonstrate that fat cells can synthesize cathelicidin, an essential molecule in the defense against *Staphylococcus aureus* infections. The study showed that dermal white AT (dWAT) expands and that preadipocytes differentiate into mature adipocytes to combat infection [28]. One key difference between dWAT and SAT is that dWAT fluctuates in size, whereas SAT’s mass is more stable [29]. The main questions raised by Zhang et al. include whether adipocytes detect bacteria directly through pattern recognition receptors or indirectly via inflammatory signals; what mechanisms trigger cathelicidin production; why adipocyte-produced cathelicidin is larger and processed differently than in other cells; and whether this isoform has distinct functions [29].

Dermis adipocytes and fibroblasts’ ability to defend against infection by producing cathelicidin declines with age. Activation of the transforming growth factor β signaling pathway reduces fibroblasts’ differentiation into adipocytes and decreases cathelicidin production, but inhibition of this pathway restored adipogenic function in adult mice [30]. Furthermore, *Candida albicans* infection triggered reactive adipogenesis in the mice’s dermis [31], where dermal fibroblasts differentiated into cathelicidin-producing preadipocytes, which helped inhibit the growth of *C. albicans*. Inhibition of adipogenesis reduced the skin’s resistance to infection, highlighting the role of adipocytes in local immunity [31]. Although these studies contributed to the clarification of the questions raised in Zhang’s study, the exact mechanisms by which adipocytes recognize pathogens remain unclear.

### 3.1. Animal Studies

Animal studies have demonstrated the expression of *Camp* in both SAT and VAT and investigated its activation pathway and role in metaflammation.

Increased expression of cathelicidin via lentivirus had no significant impact on total body weight in diabetic and non-diabetic mice fed a HFD, but it was associated with a reduction in fat mass percentage and a significant increase in lean mass percentage in diabetic mice treated with this diet. In contrast, in non-diabetic mice on a high-fat diet, overproduction of cathelicidin by lentivirus did not influence fat and lean mass [32]. Weight gain was positively associated with plasma levels of cathelicidin in obese versus lean wild-type mice [33]. Furthermore, the cathelicidin remained absent in obese C57BL6 cathelicidin-null mice (*Camp*^−/−^) fed an HFD [33]. *Camp*^−/−^ mice had lower body weight and AT mass, suggesting that the presence of cathelicidin contributes to the worsening of obesity. Cathelicidin promotes the accumulation of myeloid cells (monocytes and neutrophils) in AT, leading to local inflammation [33]. *Camp*^−/−^ mice showed higher insulin sensitivity and reduced IR compared to normal mice fed a high-fat diet in epididymal AT (VAT). Although cathelicidin stimulated the accumulation of myeloid cells, no significant differences in macrophage polarization (M1 or M2) were observed [33]. Thus, blocking cathelicidin could be a therapeutic strategy to reduce inflammation and improve insulin sensitivity in patients with obesity and MetS, paving the way for new treatments targeting AT inflammation (Table 1).

Treatment of human SAT explants and adipocytes with saturated fatty acids stimulated TLR4 signaling, leading to increased NF-κB activation. Moreover, the activation of TLR2 and TLR4 on macrophages and adipocytes by agonists resulted in their upregulation, which consequently led to the increased production of numerous inflammatory mediators. These findings provided significant proof regarding the contribution of TLRs in the inflammatory response triggered by obesity [13]. A first study conducted by Höpfinger et al. [34] explored the involvement of TLRs 2 and 4 in the regulation of Camp in SAT and VAT in male mice. Camp expression was higher in VAT as compared to SAT in male mice, and mature adipocytes expressed more LL-37 than the stromal-vascular fraction. Stimulation of TLR2 with macrophage-activated lipopeptide 2 (MALP-2) significantly increased Camp expression through the STAT3 and PI3K pathways, while LPS-induced activation of TLR4 was mediated by NF-κB and STAT3. Inhibitors of NF-κB reduced the induction of Camp expression, while inhibitors of PI3K completely blocked it, highlighting the importance of these signaling pathways in the regulation of LL-37 (Table 2, Figure 2).

CTRP-3 (C1q/TNF-related protein-3), considered a TLR4 antagonist, is an adipokine that inhibits inflammation in adipocytes and monocytes by blocking LPS-induced inflammatory pathways [38]. It is thought to counterbalance the pro-inflammatory state seen in obesity. Karrasch T. and Höpfinger A. [35] investigated how the adipokine CTRP-3 regulates CAMP expression in adipocytes via TLRs. In an in vitro model, CTRP-3 inhibited CAMP expression in adipocytes stimulated by TLR4 and TLR1/2 agonists, without affecting TLR3- or TLR2/6-induced expression, suggesting selectivity. In C57BL6 wild-type mice, using a systemic inflammatory response syndrome model, CTRP-3 inhibited LPS-induced Camp expression in a dose-dependent manner, indicating an anti-inflammatory role. CTRP-3 deficiency did not affect the basal expression of Camp or TLR genes, demonstrating that CTRP-3 regulates Camp expression only under inflammatory conditions. The study suggested that CTRP-3 could be a therapeutic target for reducing inflammation in obesity and metabolic diseases. Exploring the interplay between CTRP-3 and Camp expression made significant contributions, complementing previous findings on Camp regulation by TLR2 and TLR4. Researchers provided new insights into the role of adipokines in modulating inflammation and immune defense, identifying CTRP-3 as a crucial factor in regulating inflammatory responses in adipocytes [35].

A more recent study by Höpfinger [36] brought new insights into the role of TLRs in the regulation of Camp expression in AT. Differences in Camp expression under inflammatory conditions were studied in two types of AT: subcutaneous and intra-abdominal (visceral). TLR9 knockout mice and wild-type mice were used as experimental models to assess the TLR9-dependent effects on systemic cathelicidin concentrations. Murine 3T3-L1 adipocytes, and human SGBS adipocytes (Simpson—Golabi—Behmel syndrome adipocytes) were differentiated and stimulated with TNFα and cell-free DNA(cfDNA) in vitro. In mature human adipocytes, cathelicidin was induced by TNF-α in adipocytes through the PI3K signaling pathways, indicating the influence of TNF-α-mediated inflammation on immune responses in AT. cfDNA reduced Camp expression, suggesting that it may negatively modulate Camp under inflammatory conditions by acting on TLR7/9 receptors, in both murine and human adipocytes. A positive correlation between TLR9 and cathelicidin was observed in the SAT, but this link was not present in intra-abdominal AT, highlighting functional differences between adipose compartments in the murine model, as well as in the subcutaneous and visceral adipose tissue samples obtained from obese individuals undergoing bariatric surgery. The study revealed that PI3K plays a crucial role in TNF-induced Camp expression—a detail not previously explored in previous studies. An important finding of this study was the functional difference between SAT and VAT in terms of Camp expression. The study found a positive correlation between TLR9 and cathelicidin expression only in SAT, adding a new nuance to how these compartments respond to inflammation and infection. These findings suggest that antimicrobial peptides such as LL-37 are regulated by specific inflammatory mechanisms, contributing to a better understanding of adipose inflammation in obesity and MetS. This significantly enhances the comprehension of the interplay between immune responses and inflammatory signals, providing an additional dimension to metaflammation research.

### 3.2. Human Studies

Yang [37] conducted a study to evaluate the effects of the antimicrobial peptide LL-37 on human subcutaneous AT-derived stromal/adipose cells (ASCs) from healthy participants. LL-37 stimulated the FPRL1 receptor in a dose-dependent manner, enhancing the proliferation and migration of ASCs. LL-37 activated the MAPK pathway, which is essential for cell migration and proliferation. In addition, it stimulated the release of growth factors (vascular endothelial growth factor (VEGF), stromal cell-derived factor 1 (SDF-1), and monocyte chemoattractant protein 1 (MCP-1), which play crucial roles in blood vessel regeneration, immune cell attraction, and tissue repair. Although the study did not directly focus on MetS, the results regarding the scWAT’s inflammation pathways suggested indirect implications for the role of LL-37 in metabolic health. LL-37 may significantly contribute to SAT’s role in preventing the expansion of AT through the recruitment of new APCs. Hence, LL-37 may restore the balance in AT by guarding against hypertrophic expansion, thereby improving metabolic outcomes. However, under conditions of chronic inflammation, as is the case in hypertrophic obesity, LL-37 may contribute to such a pro-inflammatory environment that inhibits adipogenesis and promotes IR.

Cross-sectional studies have suggested a positive association between BMI and the serum level of LL-37 [39]. A study conducted in Japanese patients with type 2 diabetes did not find a significant association between LL-37 levels and BMI [40].

In a study of a cohort of normal weight and obese volunteers, BMI and skinfold thickness did not show a significant correlation with cathelicidin. In patients with BMI ≥ 25 kg/m^2^, cathelicidin was negatively correlated with insulin (*p* = 0.049) and dipeptidyl-peptidase-4 (DPP-4) (*p* = 0.004). In the entire study population, cathelicidin concentrations were negatively correlated with adiponectin, although this correlation did not reach statistical significance in subgroup analyses of patients with obesity or normal weight. In patients with a BMI ≥ 25 kg/m^2^ (*p* = 0.004) and those with a waist–hip ratio greater than 0.84, systemic concentrations of DPP-4 were positively associated with those of cathelicidin [41]. This may indicate that increased levels of insulin and DPP-4 in IR or inflammation could adversely affect cathelicidin’s synthesis or function. Furthermore, increased DPP-4 levels are linked to inflammation in AT among obese people, contributing to metabolic dysfunction. LL-37 may function as a mediator that inhibits chronic inflammation. This may lead to reduced DPP-4 activity, which plays a role in sustaining and enhancing inflammatory processes, especially in conditions such as diabetes or obesity.

A case-control study performed on 50 patients showed reduced LL-37 mRNA expression in the white blood cells of patients with MetS in comparison to the control group (*p* < 0.0001) [42]. However, BMI did not show a significant correlation with LL37 mRNA (*p* = 0.64).

Cross-sectional studies investigating the relationship between cathelicidin serum levels and BMI or weight have provided inconclusive results; however, research on the association with AT compartments has provided a deeper understanding.

A study focusing on evaluating the influence of body composition on serum levels of cathelicidin in elderly women with unipolar depression [43], showed that they had significantly higher serum levels of LL-37 compared to the control group. No significant correlations were observed between LL-37 and most body composition parameters, except for visceral fat mass, which was associated with higher LL-37 levels in women with depression. These results suggest that visceral fat may contribute to increased LL-37 production and chronic inflammation associated with depression [43].

A study conducted on the participants from the STANISLAS cohort [39] investigated the expression of the cathelicidin gene LL-37 in PBMCs and its association with cardiovascular risk factors in 90 men and 87 women. LL-37 expression in women showed a positive correlation with BMI, waist circumference, triglyceride levels, and systolic blood pressure, while HDL cholesterol levels showed a negative correlation. In men, LL-37 was positively associated with waist-to-hip ratio, fasting glucose levels, ALT activity, and neutrophil count but negatively correlated with lymphocyte count, HDL cholesterol, and apolipoprotein A-I (apoA-I). Multiple regression studies indicated that, even when controlling for waist circumference and BMI, LL-37 retained a substantial correlation with blood pressure and HDL cholesterol levels. These results confirm the association between obesity and increased levels of cathelicidin, as well as the possibility of an unfavorable metabolic profile, which is specific to MetS [39]. Hochberg et al. [44] conducted a detailed exploration of the mechanisms by which LL-37 influences immune and metabolic functions, particularly in the context of obesity and MetS; they examined the link between chronic low-grade inflammation associated with obesity and the role of cathelicidin in regulating inflammatory and immune responses, as well as the effects of LL-37 on energy metabolism and homeostasis. The study showed for the first time how metabolic changes, such as increased bile acids after bariatric surgery, can modulate LL-37 levels in adipocytes. Hochberg investigated the serum levels and expression of the CAMP gene in AT in obese individuals before and after weight loss through a low-calorie diet and bariatric surgery. These data emphasize that weight loss through bariatric surgery has beneficial effects on cathelicidin levels, particularly due to metabolic changes that occur after the intervention, such as increased bile acids, and not just due to decreased fat mass. It has been found that metabolic factors such as insulin and glucose can negatively influence cathelicidin expression in adipocytes, suggesting a link between obesity, IR, and the reduced ability of AT to participate in the defense against infections. The study demonstrated that cathelicidin expression is higher in VAT than SAT, which may have implications for understanding how different fat compartments contribute to inflammation and the immune response in obesity [44].

Cathelicidin is expressed in VAT and SAT, although there are functional disparities in its expression between these compartments. In animal studies, its expression is modulated by TLR 2/TLR 4 receptors on the PI3K/AKT and STAT3 pathways, as well as by molecules associated with subclinical inflammation (TNF-α). It may induce inflammation in AT and promote IR through these mechanisms. In humans, its expression in VAT is correlated with a worse metabolic profile, however, the exact mechanisms through which it exerts these effects remain largely unknown. Despite its synthesis being contingent upon vitamin D, the mechanisms by which it influences AT remain unclear.

## 4. Cathelicidin and Atherogenic Dyslipidemia in MetS

Dyslipidemia in MetS is primarily driven by IR, which disrupts the normal balance between lipogenesis and lipolysis, resulting in an atherogenic lipid profile characterized by elevated serum concentration of triglycerides and LDL cholesterol, along with a decrease in HDL cholesterol. Recent research has shown an association between cathelicidin serum levels and changes in the lipid profile associated with IR.

A study in pediatric patients with systemic lupus erythematosus found a negative relationship between LL-37 plasma levels and HDL and LDL cholesterol. Despite the lack of correlation between LL-37 and vitamin D (25OHD), these results indicate that LL-37 could influence lipid metabolism [45].

Research conducted by Meguro et al. [40] examined the plasma levels of LL-37 and their relationship with cardiovascular risk factors in Japanese patients with type 2 diabetes, finding a strong negative relation between LL-37 levels and HDL cholesterol, as well as positive correlations with triglycerides and high-sensitivity C-reactive protein, suggesting an association between LL-37, inflammation, and lipid profiles. As such, LL-37 plasma levels are linked to inflammatory markers and negatively associated with HDL-cholesterol in patients with type 2 diabetes [40]. However, in a group of patients with MetS, LL-37 mRNA expression was not significantly related to HDL-cholesterol (*p* = 0.76), serum total cholesterol (*p* = 0.48), or serum triglycerides (*p* = 0.215) [42].

In a cohort of normal-weight and overweight volunteers, basal serum levels of cathelicidin and LDL cholesterol showed a positive link, while HDL cholesterol and adiponectin showed a negative correlation [46]. In subgroups of women, patients with BMI <25 kg/m^2^, and non-smokers, cathelicidin concentrations were negatively correlated with HDL cholesterol (*p* < 0.001). The patients with normal weight and low skinfold thickness showed a positive correlation between LDL cholesterol and serum levels of cathelicidin [46].

The effects of oral lipid ingestion on serum levels of cathelicidin were examined, with blood samples collected before and at 2, 4, and 6 h post-ingestion. After lipid ingestion, cathelicidin serum levels significantly decreased after 4 and 6 h, alongside increased bile acid and triglyceride levels [46]. These findings suggest that the reduction in systemic serum levels of cathelicidin reduction after lipid ingestion may indicate its role as a cardiovascular biomarker linked to metabolic and AT inflammation. After lipid ingestion, no significant links with metabolic factors (BMI, skinfold thickness, triglycerides, cholesterol, glucose, or insulin) or changes in metabolic parameters were found, but a drop in serum levels of cathelicidin was associated with increased DPP-4 levels [46]. The correlation between reduced serum levels of cathelicidin and elevated DPP-4 indicates a potential connection between cathelicidin and metabolic inflammation related to fat intake, albeit in the absence of definitive associations with standard metabolic markers. The decline in cathelicidin levels following lipid intake, correlated with elevated bile acids and triglycerides, indicates that it may serve as a significant cardiovascular biomarker linked to metabolic and AT inflammation. Bile acids and triglycerides typically increase following fat consumption and are correlated with cardiovascular risk; thus, reduced serum levels of cathelicidin may signify inflammatory activation in these scenarios [46].

A significant correlation was observed between serum LL-37 and HDL cholesterol in patients with hypercholesterolemia indicating a possible role of LL-37 in regulating cholesterol [47]. The findings of this study were consistent with the results of Benachour’s research, indicating elevated expression of the LL-37 gene in PBMCs in women with obesity [39], although they diverged from the findings of Zhao’s work [48], which involved patients with myocardial infarction, explaining the discrepancies between the studies. A recent human study revealed that plasma levels of LL-37 were not significantly affected by coronary artery disease; instead, they demonstrated a correlation with leukocyte counts and proinflammatory cytokines, suggesting a link between inflammation and cathelicidin [11].

Bedi [49] suggested that IL-18 may stimulate LL-37 expression by inducing IFN-γ, increasing cathelicidin concentration, and thus influencing LDL cholesterol metabolism and the development of atherosclerosis [47,50]. The administration of atorvastatin to patients with hypercholesterolemia was associated with decreased LL-37 mRNA expression in leucocytes [47]. Ma et al. [51] found that atorvastatin can act as an anti-inflammatory therapy by downregulating IL-1β. Thus, this downregulation may cause a decrease in the circulating levels of LL-37. However, after atorvastatin therapy, a paradoxical decrease in HDL cholesterol was observed, contradicting other studies that identified an inverse association between LL-37 and HDLc in patients with T2DM [40].

Li’s study [47] suggested that the level of total cholesterol influences the level of LL-37, although other research has proposed alternative processes. LL-37 inhibited the receptor CD36 and lipid accumulation in adipocytes and hepatocytes, leading to hepatic steatosis in vivo and modulating the development of obesity [32] (Table 3).

LL-37 plays a significant role in the atherosclerotic process. In atherosclerotic lesions, researchers observed a six-fold increase in the expression of the LL-37 transcript compared to normal arteries, indicating its increased activity in plaques [50]. Apo A1 interacts with LL-37, functioning as a scavenger at the endothelial cell level, thereby safeguarding the cells from its potentially deleterious effects—a mechanism that could potentially aid in the prevention of CVD linked to endothelial dysfunction [6,52] (Table 3).

Increased concentrations of the complex formed between mitochondrial DNA and the antimicrobial peptide LL-37 (LL-37—mtDNA) were identified in atherosclerotic plaques and plasma, which showed resistance to degradation by DNase II and escaped autophagic detection. This complex activates TLR9, triggering an inflammatory response. Animal model studies have shown that LL-37—mtDNA accelerates the development of atherosclerotic lesions in apolipoprotein-E-deficient mice, and treatments with antibodies against the complex ameliorated these injuries. The findings indicate that the LL-37—mtDNA complex is important in the formation of atherosclerosis and may serve as a promising therapeutic target [53] (Table 3).

Immunohistochemical analyses have revealed that macrophages and some endothelial cells primarily produce LL-37 in the plaques, thereby confirming the active presence of the peptide in the affected areas. Additionally, treating endothelial cells with LL-37 increased the production of intercellular adhesion molecule-1 (ICAM-1) and the chemokine MCP-1, which are important for the recruitment of monocytes and the formation of atherosclerotic plaques [50] (Table 3).

Cathelicidin may act as an autoantigen involved in the immune response associated with atherosclerosis, with a variable impact depending on the administered dose. Immunization of *ApoE(*^−/−^) mice with different doses of Camp had different effects: low doses reduced the progression of atherosclerosis, whereas high doses worsened the disease by increasing the infiltration of neutrophils into atherosclerotic plaques [54].

One study [55] examined obesity-induced endothelial dysfunction and the role of NETs in this process using a murine model of diet-induced obesity. The researchers investigated whether the inhibition or degradation of NETs could mitigate endothelial dysfunction independent of weight loss or enhanced glucose management. The study found that LL-37, present in NETs, contributes to inflammation and endothelial dysfunction in obesity. Inhibiting LL-37/Camp-containing NETs prevented endothelial dysfunction, suggesting that LL-37 may be a therapeutic target for obesity-related vascular diseases.

Senescent endothelial cells may promote atherosclerosis through their proinflammatory phenotype and by amplifying inflammatory responses to atherogenic factors such as LPS and LL-37 [56]. Senescent endothelial cells show increased expression of ICAM-1 and increased activation of the NF-κB p65 pathway compared to non-senescent cells. Furthermore, receptors for LPS (TLR4) and LL-37 (P2X7) were upregulated in senescent cells, suggesting that these receptors’ activation contributes to increased inflammation [56] (Table 3).

Using human endothelial cells (HUVECs), LL-37 was shown to induce autophagy by increasing the expression of the autophagy marker LC3-II and the formation of LC3-positive puncta. LL-37 has been linked to the p62 protein, which plays a role in identifying and transporting ubiquitinated proteins to autophagosomes. However, under conditions in which autophagy is dysfunctional (through knockdown of the atg7 gene), LL-37′s degradation was delayed, and LL-37 induced cell death, accompanied by the formation of protein aggregates inside the cells [57]. Thus, although LL-37 stimulates autophagy under normal conditions, in cases of autophagic dysfunction, it may exacerbate cell death, thereby contributing to the progression of atherosclerosis. Therefore, LL-37 may harm dysfunctional endothelial cells, exacerbating inflammation and vascular damage in atherosclerosis [57].

LL-37 can be found in atherosclerotic lesions, particularly among macrophages, and has a cytotoxic effect on smooth muscle cells [58]. Exposure to LL-37 induces cellular changes associated with apoptosis, such as cell shrinkage, DNA fragmentation, and elevated caspase-3 activity, signifying LL-37-mediated cell death [58]. A subset of cells exhibited increased membrane permeability, subsequently leading to apoptosis. Nevertheless, neutrophil granulocytes showed resistance to the effects of LL-37. The findings indicated that LL-37 facilitates the apoptosis of smooth muscle cells in arterial walls, potentially compromising atherosclerotic plaques and exacerbating disease progression. Furthermore, LL-37 acts as an immune-mediated modulator in cardiovascular diseases, exacerbating inflammation in atherosclerosis by activating endothelial cells and amplifying the immune response [58] (Table 3).

## 5. Cathelicidin in Diabetes

In diabetes, IR, reduced functional β-cell mass, and high levels of FFA in the blood work together with hyperglycemia, elevated insulin levels, fat accumulation, oxidative stress, and liver inflammation [59]. Obesity accelerates this condition through genetic and epigenetic vulnerabilities, changes in the cellular microenvironment, and the microbiome-gut–brain axis. Obesogenic and diabetogenic environmental factors exacerbate metabolic imbalance through inflammation and dysfunctional autophagy, accelerating β-cell damage and elevated blood glucose. VAT expansion is associated with an altered microenvironment characterized by inflammation, fibrosis, hypoxia, imbalanced adipokine secretion, and impaired mitochondrial function. These changes prevent proper insulin signaling, induce IR, reduce insulin-stimulated glucose transport, and accelerate β-cell dysfunction, playing a central role in the development of T2DM. The obesity-induced transition from IR to T2DM involves dysfunction of both α- and β-cells in the pancreas, leading to increased expression of gluconeogenic genes in the liver. Cytokines and adipocytokines released in this context of systemic inflammation inhibit the action of insulin [60]. Inter-organ crosstalk refers to the global effects of factors secreted by a certain tissue, which can trigger physiological responses in one or more other tissues, influencing the metabolic balance or the progression of some diseases. This communication is regulated by metabolites, traditional hormones, and organokines [61]. Although not classified as an organokine, cathelicidin has extensive immunomodulatory effects, influencing inflammatory signaling, cell recruitment, and metabolic immune responses [62].

Few studies have examined the role of cathelicidin in the pathogenesis of T2DM, although LL-37 may help to control blood sugar levels, support the health of insulin-producing cells, and address COVID-19 complications linked to diabetes and hyperglycemia [63]. T2DM patients have lower LL-37 gene expression [64], and lower concentrations of LL-37 than healthy individuals [65,66,67]. In diabetes, LL-37’s expression and function can be affected by high blood glucose levels, which can compromise the immune response [66]. Serum cathelicidin levels were higher in nondiabetic and prediabetic obese patients with obesity than in nondiabetic patients with a normal body mass index [32]. Postprandial regulation of CAMP was investigated during oral glucose tolerance tests in metabolically healthy subjects [41]. The results showed a significant increase in circulating levels of cathelicidin following the ingestion of glucose, in contrast to a decrease in fatty acid binding proteins FABP2, FABP4, and DPP4. However, in SGBS adipocytes, glucose, insulin, and incretins did not significantly influence CAMP gene expression, suggesting that postprandial glucose regulates CAMP systemically but independently of adipocyte activity [41].

Metabolic-associated fatty liver disease is often associated with IR and T2DM, with excessive FFA production thought to disrupt insulin signaling pathways. In Tran et al.’s study [32], in an animal model using HFD-treated c57BL/6J wild-type and cathelicidin-deficient *Camp^−/−^* mice, one of the key findings was that cathelicidin overexpression significantly reduced hepatic steatosis in non-diabetic mice. Furthermore, the overexpression of hepatic cathelicidin diminished the accumulation of lipids in the livers of diabetic mice subjected to an HFD through the suppression of CD36 [32]. One of the major findings of this study was the effect of cathelicidin on lowering CD36 expression in adipocytes and hepatocytes, suggesting its potential for alleviating steatosis. Excess fatty acids contribute to insulin resistance; hence, the suppression of CD36 by cathelicidin may minimize lipotoxicity and subsequently increase insulin signaling.

In T2DM, oxidative stress induced by hyperglycemia and hyperlipidemia promotes the generation of ROS. This inhibits Ca^2+^ mobilization and activates proapoptotic signals in the pancreatic β-cells [68]. Results from animal studies showed a protective effect of cathelicidin on β-cell survival [69]. In the INS-1 832/13 rat insulinoma clone, Camp was constitutively expressed by β cells. Butyrate and phenyl butyric acid induce Camp expression, and inflammatory stimuli such as IL-1 or LPS can trigger its secretion. Camp promotes β-cell survival by modulating anti-apoptotic proteins via EGFR and stimulating insulin secretion. In diabetes-prone mice, Camp enhances glucose-stimulated insulin secretion and modulates inflammatory responses, reducing β-cell apoptosis. *Camp*-deficient mice exhibit defective insulin secretion, while Camp administration improves glucose clearance in prediabetic mice. This study’s findings indicate that cathelicidins positively influence β-cell activity and may serve as a potential therapeutic target for disorders associated with β-cell malfunction [69].

The amyloid self-assembly of islet amyloid polypeptide (IAPP) is associated with pancreatic inflammation, β-cell degeneration, and the pathophysiology of T2DM [70,71]. Recent research indicates that the human cathelicidin LL-37 binds to IAPP with great affinity, effectively blocking its amyloid production and reducing β-cell damage in vitro. The findings indicate that LL-37 plays a protective role in T2DM, offering a molecular foundation for the development of LL-37-derived peptides with antibacterial, immunomodulatory, and anti-amyloid characteristics, positioning them as viable candidates for multifunctional treatments [70].

T2DM is characterized by chronic low-grade inflammation, and LL-37 plays an immunomodulatory role in regulating inflammation. However, in the presence of high glucose levels, inflammatory modulation of LL-37 can be dysregulated, contributing to immune dysfunction and chronic inflammation, which exacerbate complications of diabetes, including cardiovascular ones [72]. Conversely, despite being limited in quantity, current research indicates that the beneficial impact of LL-37 on type 2 diabetes mellitus is attributed to its modulation of insulin resistance, along with its protective influence on pancreatic beta cell mass through its antiapoptotic properties and reduction in pancreatic amyloid synthesis.

## 6. Cathelicidin and Lifestyle Components

Reduced antimicrobial defenses in the skin associated with obesity increase the risk of infection. A study in mice investigated how diet-induced obesity affects the immune function of adipocyte progenitor cells in the dermis. Using a high-fat diet to induce obesity, researchers found that obese mice had a decreased ability to produce antimicrobial peptides such as cathelicidin, impairing their skin’s defense against infections. Additionally, the depletion of adipocyte progenitors and increased TGF β levels, which suppress immune function, were observed. The study demonstrated that obesity negatively impacts the skin’s immunity, contributing to a higher risk of infections [73].

Another study in mice involved using exercise to test the inflammatory and immune response to infection with drug-resistant *Staphylococcus aureus* [74]. Regular exercise increased cathelicidin levels in the VAT and lungs, being associated with a reduction in adipocyte size. Exercise also reduced serum the levels of TNF-α and IL-1β, while significantly decreasing NO production in the lungs.

Research by Li et al. [75] analyzed the impact of cathelicidin deficiency on liver damage induced by a high-fat diet combined with alcohol in a murine model. Cathelicidin plays a central role in the regulation of lipid metabolism, inflammation, and liver injury. Cathelicidin deficiency could protect against the adverse effects of a high-fat and high-alcohol diet by modulating FGF21/adiponectin signaling and lipolysis in white AT. The implications for obesity and MetS are significant, suggesting that cathelicidin levels may directly influence how the body manages fat accumulation and inflammation, two major components of MetS.

Proteomic analyses from participants in the OmniHeart randomized trial and the Atherosclerosis Risk in Communities Study [76] indicated that the antimicrobial peptide cathelicidin is one of nine proteins that facilitate the relationship between lipoprotein levels and healthy diets, with varying macronutrient proportions. Cathelicidin was identified as having a positive association with lipoproteins, including HDL cholesterol and triglycerides, suggesting an important role in the regulation of lipid metabolism. This finding highlights the involvement of cathelicidin not only in the immune response but also in metabolic processes that influence cardiovascular disease risk, reinforcing the idea that inflammatory and antimicrobial proteins play a role in cardiovascular health.

## 7. Conclusions

Cathelicidin, an antimicrobial and immunomodulatory peptide, plays a crucial role in the innate immune response while also having important implications for MetS. Recent research suggests that cathelicidin is involved in chronic inflammatory processes central to the pathogenesis of MetS, including obesity, IR, and T2DM. The persistent inflammation characteristic of this syndrome is exacerbated by AT accumulation and metabolic dysfunction, and cathelicidin can modulate this response through its anti-inflammatory and tissue regeneration-promoting effects. In addition, cathelicidin has been associated with protection against infections and cardiovascular complications, both of which are common in the setting of metabolic syndrome. Thus, cathelicidin may serve as a marker of the constellation of metabolic disturbances associated with MetS.

## Figures and Tables

**Figure 1 metabolites-14-00672-f001:**
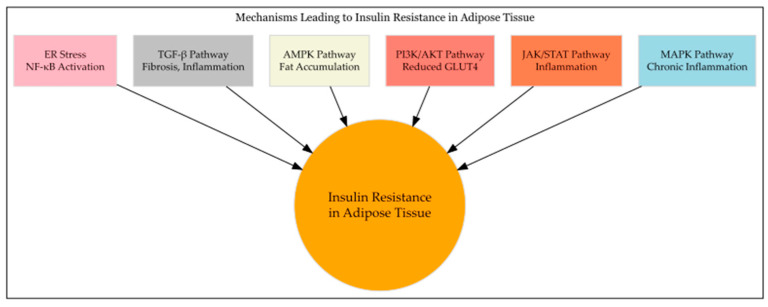
Mechanisms of IR—cellular pathways contributing to the development of IR. Legend: Each pathway affects specific aspects of metabolism, inflammation, or cellular stress, all culminating in IR, a hallmark of MetS and type 2 diabetes.

**Figure 2 metabolites-14-00672-f002:**
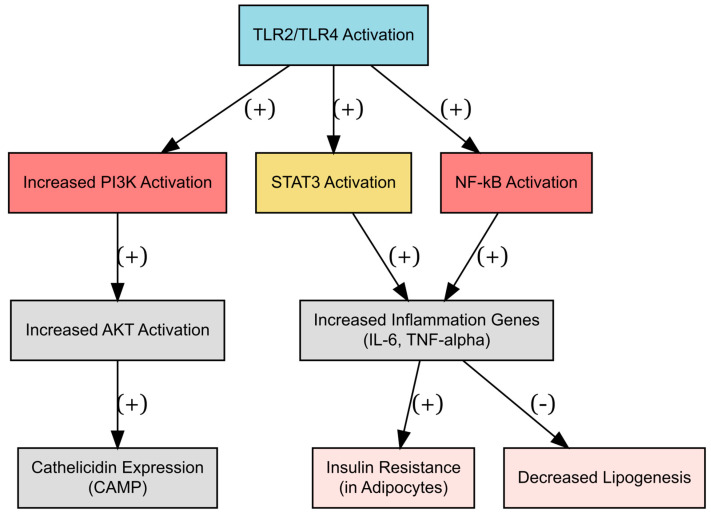
The role of Toll-like receptors in AT and cathelicidin expression. Legend: TLR2/TLR4 activation drives a cascade of events leading to inflammation, cathelicidin expression, and metabolic dysfunction. The activation of the PI3K/AKT, STAT3, and NF-κB pathways collectively upregulates pro-inflammatory genes, promoting IR in adipocytes and decreasing the formation of new fat cells (lipogenesis). These processes are central to the development of metabolic diseases such as obesity and T2DM, where chronic inflammation disrupts normal metabolic homeostasis.

**Table 1 metabolites-14-00672-t001:** Role of cathelicidin in adipose tissue inflammation (animal studies).

References	Adipose Tissue Type	Induction Time/Study Model	Observed Effects in AT
Zhang et al., 2015 [28]	VAT	KO specific to AT	↓ Secretion of proinflammatory cytokines
Braster et al., 2016 [33]	Epididymal AT (VAT) and SAT	*Camp*^−/−^ mice on HFD for 18 weeks	↑ Insulin sensitivity in *Camp*^−/−^ mice; ↓ Weight gain; ↓VAT inflammation
Höpfinger, 2021 [34]	Gonadal (visceral) AT and SAT	Wildtype male and female C57BL/6N mice	↑ Camp expression: in VAT. in mature adipocytes compared to SVF.
Karrasch, 2021 [35]	Visceral and SAT	Systemic inflammatory response model in mice (LPS) C57BL/6 Mice; CTRP-3 KO mice)	CTRP-3: ↓ LPS-induced Camp expression in AT. selectively regulates Camp only under inflammatory conditions.
Höpfinger et al., 2023 [36]	Visceral AT and SAT	TLR9 knockout (KO) and wildtype C57BL/6N mice, with Camp expression analyzed on SAT and VAT	↓ LL-37 in TLR9 KO mice No significant differences in Camp expression in adipocytes between TLR9-KO and wildtype mice.
Molhoek et al., 2009 [8]	N/A	KO in CD4+ T cells	Limited impact on chronic inflammation in obesity
Tran et al. (2016) [32]	Liver, mesenteric fat	High-fat diet + diabetes in C57BL/6J mice	Cathelicidin overexpression: ↓ hepatic steatosis, and fat mass; CD36 and lipid accumulation in adipocytes and hepatocytes

**Table 2 metabolites-14-00672-t002:** Role of cathelicidin in adipose tissue inflammation (in vitro studies).

References	In Vitro Study	Induction Time/Study Model	Observed Effects in AT
Höpfinger, 2021 [34]	In vitro (3T3-L1 and SGBS adipocytes)	Differentiated 3T3-L1 murine adipocytes, stimulated with a TLR2 agonist (MALP-2) and TLR4 agonist (LPS).	↑ Camp by TLR2 and TLR4 activation;
Karrasch, 2021 [35]	3T3-L1 and human SGBS adipocytes	Differentiated 3T3-L1 murine adipocytes and human SGBS adipocytes, stimulated with TLR4 and TLR1/2 agonists (LPS and Pam3CSK4 respectively).	CTRP-3 ↓ Camp expression induced by TLR4 and TLR1/2 agonists, without affecting expression induced by TLR3 or TLR2/6 agonists.
Höpfinger et al., 2023 [36]	3T3-L1 and human SGBS adipocytes	TLR9 KO model in mice	TNFα ↑ CAMP expression in mature adipocytes. cfDNA ↓ Camp expression in murine and human adipocyte cultures.
Yang et al., 2016 [37]	Human	Human ASCs isolated from healthy donors, stimulated with LL-37.	LL-37 ↑ ASC proliferation and migration (+MAPK pathway, via EGR1).
Tran et al. (2016) [32]	3T3-L1 preadipocytes, human mesenteric fat preadipocytes	In vitro on 3T3-L1 cells, human mesenteric fat tissues obtained during surgery	↑ Cathelicidin → ↓ lipid accumulation in murine and human-derived adipocytes;

**Table 3 metabolites-14-00672-t003:** Role of cathelicidin in atherosclerosis.

Author (Year)	Study Type	Tissue	Study Model	Observed Effects
Tran et al. (2016) [32]	Animal	Liver, mesenteric fat	High-fat diet + diabetes in C57BL/6J mice	↑↑ Cathelicidin: ↓ Hepatic steatosis, mesenteric fat, and fat mass; ↓ CD36 and lipid accumulation
Edfeldt et al. (2006) [50]	In vitro	Atherosclerotic plaques	Human endothelial cells, macrophages	LL-37 ↑ in atherosclerotic lesions by macrophages; induced MCP-1 and ICAM-1 → inflammation in vitro
Svensson et al. (2017) [52]	In vitro	Endothelial cells (HUVECs)	Human umbilical vein endothelial cells	ApoA-I binds to LL-37 ↓ LL-37-induced cytotoxicity ↑ Cell viability by counteracting endothelial dysfunction
Zhang et al. (2015) [53]	Animal, in vitro	Atherosclerotic lesions	ApoE-deficient mice, HUVEC	LL-37-mtDNA complex escaped autophagy and DNase II degradation → ↑ TLR9, ↑ atherosclerosis in mice
Mihailovic et al. (2017) [54]	Animal, in vitro	Aortic sinus, spleen	ApoE(-/-) mice, in vitro on splenocytes	Camp as a self-antigen ↑ CD8+ T cell; ↑ Atherosclerosis at high doses—effect varied by diet (normal vs. high fat)
Wang et al. (2018) [55]	Animal	Mesenteric arterioles	Diet-induced obesity in C57BL/6J mice	NETs formed in obesity ↑ Camp in arterial walls → ↓ vasorelaxation;
Suzuki et al. (2019) [56]	In vitro	Senescent endothelial cells	Serially passaged HUVECs	LL-37 and LPS ↑ ICAM-1 and NF-κB p65 phosphorylation in senescent endothelial cells → ↑TLR4 and P2X7 expression
Suzuki et al. (2022) [57]	In vitro	Endothelial cells (HUVECs, HCAECs)	Atg7 knockdown cells, lysosomal inhibitor-treated HUVECs	LL-37 → autophagy, ↑ cell death in autophagy → dysfunctional cells by ↑ protein aggregates
Ciornei et al. (2016) [58]	In vitro	Vascular smooth muscle cells	Human atherosclerotic plaques and cultured smooth muscle cells	LL-37 ↑ apoptosis-like death in vascular smooth muscle cells (DNA fragmentation, caspase-3 activation, and LDH leakage)

## Data Availability

No new data were created or analyzed in this study. Data sharing is not applicable to this article.

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
