# Peer review of "Cathelicidin: Insights into Its Impact on Metabolic Syndrome and Chronic Inflammation"

_metabolites, 2024, doi:10.3390/metabo14120672_

Round 1
Reviewer 1 Report
Comments and Suggestions for Authors
The review “Cathelicidin: Insights into Its Impact on Metabolic Syndrome and Chronic Inflammation“ focuses on LL-37, a human cathelicidin associated with metabolic syndrome (MetS). Via analysis of the relationship between cathelicidin and components of MetS it has been concluded that LL-37 is associated with inflammation in adipose tissue and IR development in obesity. Also, LL-37 was noted to affect the lipid metabolism, with a negative relationship between its levels and HDL cholesterol. Thus, it has been concluded that LL-37 is essential for regulating the balance between inflammatory and anti-inflammatory responses in MetS. The review features unusual structure. Like a research paper, it encompasses chapters, which are typical for these submissions: Introduction, Materials and methods, Results, Conclusions.
1) I would recommend sticking to a more typical structure for a review.
2) Another feature: too many abbreviations. Some of them are introduced but not used, e.g. ARIC in line 686. I would recommend editing them and presenting in a special section of the review, called Abbreviations.
3) There are improper citations. E.g. the sentence in lines 119-120 sounds “Their amphipathic and cationic properties drive the antimicrobial activity against Gram-positive and Gram-negative bacteria at neutral pH [10,12].” The first work, [10] deals with the mechanism of activity against Gram-negative bacteria. The second work does not cite any antibacterial properties of LL37 and deals with correlation of its level with 25-Hydroxyvitamin D level.
4) I would recommend adding a Figure illustrating amphipathic character of LL37 structure, where also amino acid sequence of this peptide is provided. This will facilitate reading.
5) The work finishes with conclusion: “Studying cathelicidin and how it regulates inflammation, and metabolic function provides valuable insights into the development of new therapeutic strategies in the management of MetS and its associated complications.” However, I do not see any details. I think, a section called “LL37 in management of MetS” would be desirable.
6) English grammar needs to be checked. Some errors are indicated below.
Minor comments:
Line 49. “CVD)” CVD. End of sentence here?
Lines 105, 109: “kD” Better kDa, like in line 74.
Figure 1. “Legend: This figure illustrates”. Please omit this phrase and be concise. Also, add what are “+/-“ symbols designate. This is also necessary to do for Figure 3. For this Figure the word “upregulates” seems to be extra, if explanations of “+” and “-“ signs have been added.
Line 187: “The 2015 study by Zhang et al. [23] was the first to demonstrate” Better: In 2015, Zhang and others demonstrated for the first time…
Line 239: “the causes of adipogenesis reduces” reduction?
Line 275: “All figures and tables should be cited in the main text as Figure 1, Table 1, etc.” What for is this phrase here?
Line 278, 377: See above notion for Figure 1.
Lines 469, 470. Please, explain what are the values r, p in brackets.
Line 475: 0.001). Missing bracket.
Line 515: “Antimicrobial peptide LL-37”. Simply, “LL37”, Omit “Antimicrobial peptide”. It has already been defined.
Line 672: “resistant”. It is necessary here “drug-resistant”, or “antibiotic-resistant”.
Author Response
Reviewer 1
The review “Cathelicidin: Insights into Its Impact on Metabolic Syndrome and Chronic Inflammation“ focuses on LL-37, a human cathelicidin associated with metabolic syndrome (MetS). Via analysis of the relationship between cathelicidin and components of MetS it has been concluded that LL-37 is associated with inflammation in adipose tissue and IR development in obesity. Also, LL-37 was noted to affect the lipid metabolism, with a negative relationship between its levels and HDL cholesterol. Thus, it has been concluded that LL-37 is essential for regulating the balance between inflammatory and anti-inflammatory responses in MetS. The review features unusual structure. Like a research paper, it encompasses chapters, which are typical for these submissions: Introduction, Materials and methods, Results, Conclusions.
- I would recommend sticking to a more typical structure for a review.
Thank you for your suggestion. We modified the structure of the article accordingly and we excluded the Material and Methods section.
- Another feature: too many abbreviations. Some of them are introduced but not used, e.g. ARIC in line 686. I would recommend editing them and presenting them in a special section of the review, called Abbreviations.
Thank you for pointing this out. We reviewed the abbreviations and excluded those that did not repeat. Furthermore, we added a table called Abbreviations before the Reference section.
- There are improper citations. E.g. the sentence in lines 119-120 sounds “Their amphipathic and cationic properties drive the antimicrobial activity against Gram-positive and Gram-negative bacteria at neutral pH [10,12].” The first work, [10] deals with the mechanism of activity against Gram-negative bacteria. The second work does not cite any antibacterial properties of LL37 and deals with correlation of its level with 25-Hydroxyvitamin D level.
This is a very important aspect. We reviewed the references and corrected them.
- I would recommend adding a Figure illustrating amphipathic character of LL37 structure, where also amino acid sequence of this peptide is provided. This will facilitate reading.
Thank you for your valuable suggestion. We included a figure depicting the amphipatic character of LL-37.
- The work finishes with conclusion: “Studying cathelicidin and how it regulates inflammation, and metabolic function provides valuable insights into the development of new therapeutic strategies in the management of MetS and its associated complications.” However, I do not see any details. I think, a section called “LL37 in management of MetS” would be desirable.
Thank you for this important commentary. After our knowledge there are not studies on targeting LL-37 as a therapeutic target in metabolic syndrome or its components. Therefore we modified the chapter’s title ‘Cathelicidin and lifestyle components - potential therapeutic role’ in ‘Cathelicidin and lifestyle components’ and the conclusions by removing the term therapeutic.
- English grammar needs to be checked. Some errors are indicated below.
We corrected this aspect by using MDPI’s English editing service.
Minor comments:
Line 49. “CVD)” CVD. End of sentence here?
Thank you – we replace ) with .
Lines 105, 109: “kD” Better kDa, like in line 74.
We replace kD with kDa.
Figure 1. “Legend: This figure illustrates”. Please omit this phrase and be concise. Also, add what are “+/-“ symbols designate. This is also necessary to do for Figure 3. For this Figure the word “upregulates” seems to be extra, if explanations of “+” and “-“ signs have been added.
We removed the following sentences: “This figure illustrates the dual function of cathelicidin (LL-37) in MetS, highlighting its involvement in infection defense and regulation of metabolic inflammation through AT mechanisms. “.
Line 187: “The 2015 study by Zhang et al. [23] was the first to demonstrate” Better: In 2015, Zhang and others demonstrated for the first time…
We modified the text as you suggested.
Line 239: “the causes of adipogenesis reduces” reduction?
Thank you for this remark. We replace reduces with reduction.
Line 275: “All figures and tables should be cited in the main text as Figure 1, Table 1, etc.” What for is this phrase here?
We removed this raw that belonged to the MDPI template.
Line 278, 377: See above notion for Figure 1.
Thank you for this important suggestion. We modified the tile of the figure and removed the sentences – ‘This figure represents the mechanisms leading to IR in AT, showing how different cellular pathways contribute to the development of IR’.
Lines 469, 470. Please, explain what are the values r, p in brackets.
The values in brackets represent two statistical measures: r and p. We remove the r value to improve the clarity of the text since the communication of the r-value does not bring additional information relevant to the subject.
Line 475: 0.001). Missing bracket.
Thank you for this remark – we modified this.
Line 515: “Antimicrobial peptide LL-37”. Simply, “LL37”, Omit “Antimicrobial peptide”. It has already been defined.
We deleted ‘Antimicrobial peptide’ from the beginning of the sentence.
Line 672: “resistant”. It is necessary here “drug-resistant”, or “antibiotic-resistant”.
We used the term drug-resistant.
Reviewer 2 Report
Comments and Suggestions for Authors
I have reviewed the manuscript “Cathelicidin: Insights into Its Impact on Metabolic Syndrome and Chronic Inflammation”. It is a review article that covers comprehensively the actual knowledge of the participation of cathelicidin on various aspects of metabolic syndrome and its relation to chronic inflammation.
In my opinion, the manuscript can be accepted after minor revision since only grammatical errors have to be corrected. For example:
1.- Line 67, change cells lymphocytes for lymphocytes cells.
2.- Give the whole name before using abbreviatures, as in line 95 in RCTs.
3.- Scientific names should be written in italics (i.e., line 188, 201, 203, 590).
4.- Delete line 275’s text.
Author Response
Reviewer 2
I have reviewed the manuscript “Cathelicidin: Insights into Its Impact on Metabolic Syndrome and Chronic Inflammation”. It is a review article that covers comprehensively the actual knowledge of the participation of cathelicidin on various aspects of metabolic syndrome and its relation to chronic inflammation.
In my opinion, the manuscript can be accepted after minor revision since only grammatical errors have to be corrected. For example:
1.- Line 67, change cells lymphocytes for lymphocytes cells.
We modified the text as you suggested.
2.- Give the whole name before using abbreviatures, as in line 95 in RCTs.
Thank you for pointing this out. We reviewed the abbreviations and we added a table called Abbreviations before the Reference section
3.- Scientific names should be written in italics (i.e., line 188, 201, 203, 590).
Thank you for this advice. We checked for the scientific names and wrote the names of genes and proteins according to the UniProt as recommended in the Submission Guidelines (https://www.uniprot.org/uniprotkb/P49913/entry ). Furthermore, we wrote in italics the names of the bacteria and fungi.
4.- Delete line 275’s text.
We removed this raw that belonged to the MDPI template. We apologize for the inconvenience.
Reviewer 3 Report
Comments and Suggestions for Authors
Popa et al. summarized the role of Cathelicidin in metabolic syndrome and chronic inflammation. Overall, I feel that this manuscript is a little broad and lacks focus in terms of the underlying mechanisms. I have several questions and comments as below:
1. It seems that the authors focus on the effect of Cathelicidin on adipose tissue. Whether Cathelicidin is mainly secreted by adipose tissue? In adipose tissue, which cell type primarily secretes Cathelicidin in adipose tissue?
2. The authors also emphasize chronic inflammation. I suggest that the authors focus on one type of immune cell, such as macrophage, to elucidate the role of Cathelicidin in chronic inflammation and the underlying molecular mechanisms. The authors mentioned a lot of immune cells, which is not helpful to provide deep insight.
3. For the diabetes section, the authors can focus on the effect of cathelicidin on insulin resistance and insulin secretion. The current description of this section is too general and random.
4. If possible, the authors can cite the articles of tissue or cell-specific knockout of cathelicidin to provide the precise mechanisms.
5. The authors used several different terms of Cathelicidin ( LL-37,CAMP). If possible, the authors had better use same term of Cathelicidin in the whole manuscript.
6. The authors should discuss the potential receptors of Cathelicidin, which will provide mechanistic insight.
7. The authors can provide a diagram to summarize the effect of Cathelicidin on metabolic syndrome and chronic inflammation.
8. The writing of this manuscript should be more accurate. When cite the research articles, the authors should indicate which tissue and what kind of physiological condition.
Author Response
Reviewer 3
Popa et al. summarized the role of Cathelicidin in metabolic syndrome and chronic inflammation. Overall, I feel that this manuscript is a little broad and lacks focus in terms of the underlying mechanisms. I have several questions and comments as below:
- It seems that the authors focus on the effect of Cathelicidin on adipose tissue. Whether Cathelicidin is mainly secreted by adipose tissue? In adipose tissue, which cell type primarily secretes Cathelicidin in adipose tissue?
Thank you for these very interesting remarks. We focused on the adipose tissue because of its crucial role in developing MetS through the metaflammation triggered by the adipocytes' hypertrophy and the secondary low-grade subclinical chronic inflammation, which contribute to insulin resistance, the central feature of MetS. This focus may seem misconducting, therefore to emphasize cathelicidin's main source and role we started with the section entitled Cathelicidin Overview and we moved the text between the lines 163-184 at the beginning of section 4 and we add the following paragraph:
“The main source of cathelicidin in humans is neutrophils (Sørensen, 2001), where it is stored as the precursor hCAP18 and subsequently cleaved extracellularly to form LL-37 during systemic immune responses (Sorensen et al., 2001). Adipose tissue contributes to cathelicidin production as a secondary or induced source, especially during infection or inflammation. Within adipose tissue, preadipocytes appear to be the main cell type responsible for cathelicidin synthesis.”
- The authors also emphasize chronic inflammation. I suggest that the authors focus on one type of immune cell, such as macrophage, to elucidate the role of Cathelicidin in chronic inflammation and the underlying molecular mechanisms. The authors mentioned a lot of immune cells, which is not helpful to provide deep insight.
Thank you for the suggestion. Macrophages play an essential role in subclinical inflammation in obesity. However, they are not the only immune cells involved in this chronic low-grade inflammation localized in the AT. This is the result of a complex network of interactions between various immune cell types, including neutrophils, CD4+ Th2 T cells, eosinophils, and macrophages. Cathelicidin does not only act on macrophages; it is active in relation to several cell types and can influence the inflammatory balance in AT. In obesity, cathelicidin can modulate the functions of neutrophils, which are among the first cells recruited in AT at the start of the inflammatory process, as well as T lymphocytes and NK cells, which maintain inflammation. We believe that an expanded insight into the role of LL-37 in different cell populations in AT would provide a more comprehensive picture of the mechanisms that maintain subclinical inflammation in obesity. Focusing exclusively on macrophages could limit our understanding of this complex interaction and would not fully capture the immunoregulatory role of cathelicidin in AT inflammatory networks.
- For the diabetes section, the authors can focus on the effect of cathelicidin on insulin resistance and insulin secretion. The current description of this section is too general and random.
This is a very important aspect; we checked for this aspect and removed the text that was non-relevant for the implication of LL-37 in the pathogenesis of T2DM.
- If possible, the authors can cite the articles of tissue or cell-specific knockout of cathelicidin to provide the precise mechanisms.
We have included tables to organize the information and checked existing references. At this time, we have not identified any additional articles on tissue- or cell-specific knockout of cathelicidin (CAMP/LL-37/CRAMP) relevant to the metabolic syndrome. If you have specific recommendations for such articles, please let us know, and we will include the additional information required
- The authors used several different terms of Cathelicidin ( LL-37,CAMP). If possible, the authors had better use same term of Cathelicidin in the whole manuscript.
Thank you for this advice. We checked for the different names and modified them according to the https://www.uniprot.org/uniprotkb/P49913/entry recommendations on the terminology of CAMP gene and its derived proteins:
“Protein names
- Recommended name
Cathelicidin antimicrobial peptide
- Alternative names
- 18 kDa cationic antimicrobial protein; (CAP-18; hCAP-18)
- Cleaved into 8 chains
- Antibacterial peptide FALL-39 - -Alternative names: FALL-39 peptide antibiotic
- Antibacterial peptide LL-37
- Antibacterial peptide KR-20
- Antibacterial peptide LL-23
- Antibacterial peptide LL-29
- More chains
Gene names
- Name
CAMP
- Synonyms
CAP18, FALL39
- ORF names
HSD26”
Furthermore, the protein coded by CAMP gene is called Cathelicidin antimicrobial peptide (CAMP or hCAP-18). The CAMP gene product is processed by a serine protease-dependent mechanism into multiple novel antimicrobial peptides distinct from and shorter than cathelicidin LL-37, such as peptides KR-20 (residues 151-170), LL-23 (residues 134-156), LL-29 (residues 134-162), KS-30 (residues 141-170), RK-31 (residues 140-170) and FF-33 (residues 138-170) (https://www.uniprot.org/uniprotkb/P49913/entry ). Therefore, we retained the protein names as they appear in the cited articles, updating the older term CRAMP to CAMP and keeping the name LL-37.
- The authors should discuss the potential receptors of Cathelicidin, which will provide mechanistic insight.
Thank you for this important suggestion. We added a new section regarding cathelicidin receptors and their potential involvement in MetS
7.The authors can provide a diagram to summarize the effect of Cathelicidin on metabolic syndrome and chronic inflammation.
Thank you for the suggestion. This is a very important aspect, and we provide a graphical abstract for the article.
- The writing of this manuscript should be more accurate. When cite the research articles, the authors should indicate which tissue and what kind of physiological condition.
This is a very important aspect, we checked for this aspect and added supplementary information regarding the type of study and tissue to improve the clarity of the manuscript.
Round 2
Reviewer 1 Report
Comments and Suggestions for Authors
The review has not been improved significantly. Although Figure 1 has been added, no details are provided. The legend to this Figure is too short. Why the structure is amphiphilic? What is shown in the Figure? It is necessary to stress that this structure is adopted by the peptide in the membrane mimetic environment (The reference 21 should be changed to: Wang G. Structures of human host defense cathelicidin LL-37 and its smallest antimicrobial peptide KR-12 in lipid micelles. J Biol Chem. 2008 Nov 21;283(47):32637-43. doi: 10.1074/jbc.M805533200. Epub 2008 Sep 25. PMID: 18818205). What is the structure of the peptide in aqueous solution? The peptide has many functions, according to the review. Does it mean that the peptide is flexible and can interact with some proteins, not only with lipid membrane? Without addressing these issues, the review looks like a combination of unrelated facts. This variant is a too raw for publishing.
Author Response
Response to reviewer 1
Thank you very much for your objective comments which were very important and helpful to improve the scientific soundness of our paper.
- Although Figure 1 has been added, no details are provided. The legend to this Figure is too short.
Thank you for pointing this out. We added the heliogram to the figure to better depict the amphipathic character of the LL-37 peptide and its related characteristics – net charge and hydrophobic moment (HMom). We extended the explanations included in the legend of the figure.
Figure 1. LL-37: Tetrameric Assembly and Helical Wheel Representation
Legend: A. Tetrameric structure of LL37 adopted by the peptide in the membrane mimetic environment (available at https://www.rcsb.org/structure/2K6O ) B. Helical Wheel Representation (available at https://heliquest.ipmc.cnrs.fr/cgi-bin/ComputParamsV2.py). The properties of the amino acids are indicated by the color code: hydrophobic (yellow), polar (blue/green), or charged (red). The vector (arrow) represents the hydrophobic moment (HMom), a measure of hydrophobic asymmetry around the helix, with a value of 0.502 (average hydrophobicity). Created in BioRender. Caba, L. (2024) https://BioRender.com/u92g42
- Why the structure is amphiphilic?
Thank you for your question. We appreciate the opportunity to clarify this aspect and we included this explanation in lines – 104-122.
Cathelicidin has three distinct domains: the signal peptide located at the N-terminal end, which plays a role in transporting the protein to its activation site where is processed by proteinase 3; the cathelin-like domain, well-conserved across species, which contributes to protein stabilization; and the antimicrobial domain located at the C-terminal end, responsible for disrupting bacterial membranes, neutralizing toxins, and participating in LL-37's functions within the immune system.
LL-37 has a sequence of amino acids that forms distinct hydrophobic and hydrophilic faces when folded into an α-helical structure. The hydrophobic regions, composed primarily of nonpolar amino acids like leucine, isoleucine, phenylalanine, and valine, interact with the lipid core of the membrane, enabling LL-37 to be inserted into bacterial membranes. Conversely, hydrophilic regions, rich in charged residues such as arginine and lysine, interact with the polar head groups of the lipids and the aqueous environment. This arrangement forms an amphipathic helix (helix-amphipathic-bent-helix, residues 2-31) when is surrounded by SDS (sodium dodecyl sulfate) micelles. One face of the helix interacts with the lipid bilayer of the cell membrane, while the opposite face is exposed to the aqueous environment. Specific amino acids in the C-terminal tail such as Phe-5, Phe-6, Phe-17, and Phe-27, along with the arginines, interact directly with the micelles, suggesting compatibility for negatively charged structures such as bacterial membranes .
- What is shown in the Figure?
It is necessary to stress that this structure is adopted by the peptide in the membrane mimetic environment (The reference 21 should be changed to: Wang G. Structures of human host defense cathelicidin LL-37 and its smallest antimicrobial peptide KR-12 in lipid micelles. J Biol Chem. 2008 Nov 21;283(47):32637-43. doi: 10.1074/jbc.M805533200. Epub 2008 Sep 25. PMID: 18818205).
We gratefully appreciate your valuable remark. We have addressed your comment by adding the requested explanation to the figure legend and including updated references to ensure clarity and provide appropriate attribution. We acknowledge your constructive feedback, which has helped improve the quality and transparency of our manuscript.
- What is the structure of the peptide in aqueous solution?
We are thankful for your suggestion and have provided this explanation in the lines 123-131.
In aqueous solutions at very low concentrations, LL-37 can exist as a monomer, dimer, or tetramer. The ability of LL-37 to form these oligomers is attributed to its positive charge and the presence of an N-terminal fragment (containing 30% hydrophobic amino acids), which acts as the hydrophobic core of the oligomer. Circular dichroism spectroscopy revealed that LL-37 and its all-D enantiomer adopted well-defined alpha-helical structures in physiological saline (PIL) and in DMEM (Dulbecco's Modified Eagle Medium) due to intramolecular salt bonds that stabilize the structure. LL-37 lacks disulfide bridges and generates stable oligomers that are resistant to proteolytic destruction.
- The peptide has many functions, according to the review. Does it mean that the peptide is flexible and can interact with some proteins, not only with lipid membrane? Without addressing these issues, the review looks like a combination of unrelated facts. This variant is a too raw for publishing.
Thank you for pointing this out. We appreciate your observation. We added the following paragraphs to provide a more detailed explanation.
While LL-37's amphipathic nature allows it to effectively interact with and disrupt bacterial cell membranes via mechanisms such as the "barrel-stave," "carpet," and "toroidal" models, its ability to exist as monomers, dimers, or tetramers in aqueous solution at low concentrations indicates a degree of structural flexibility. This ability implies flexibility and enables interactions with various biological targets, including proteins and lipid membranes. The flexibility of the peptide facilitates an “induced fit” mechanism during protein binding. This means that when the peptide approaches its target, its structure can undergo changes to better match the target protein, thereby increasing both the affinity and the specificity of the interaction. This flexible structure allows it to efficiently interact with cell membranes and various proteins involved in its immunomodulatory functions, adapting as needed to have an optimal effect.
Mechanistically, LL-37 binds to negatively charged components of bacterial membranes, oligomerizes, and forms channels that allow the translocation of water and ions, disrupting membrane integrity. Studies have shown that AMPs can target ribosomal complexes and other intracellular structures, suggesting that their interaction capabilities extend beyond mere membrane disruption. LL-37 interacts with the bacterial proteins LiaX and PhoQ, components of two-component systems, triggering the remodeling of lipopolysaccharides (LPS), a mechanism through which bacteria increase their resistance to the action of the LL-37 peptide.
Beyond its antimicrobial activity, LL-37 interacts with various cell types, inducing chemokine expression, neutralizing endotoxins, stimulating angiogenesis, and regulating apoptosis. It modulates inflammatory responses by interacting with protein structures in pro-inflammatory pathways. For example, LL-37 binds to proteins in the TLR4 receptor complex, such as Lipopolysaccharide-Binding Protein and Myeloid Differentiation factor 2 (MD2), preventing the activation of inflammatory pathways and reducing the expression of pro-inflammatory mediators like TNF-α [10]. In addition to its functions in the immune system, LL-37 is involved in metabolic regulation due to its ability to penetrate the cell membrane and interact with intracellular proteins, such as GAPDH (glyceraldehyde-3-phosphate dehydrogenase). Furthermore, LL-37 can modulate the activity of certain proteins without activating them as a traditional ligand would. For example, in adipocytes and hepatocytes, LL-37 inhibits the function of the CD36 receptor, which is involved in processes such as lipid metabolism and inflammation, demonstrating its capacity to directly alter protein activity.
Sincerely yours,
Andreea Gherasim MD, PhD, lecturer
On behalf of the authors
Reviewer 3 Report
Comments and Suggestions for Authors
No further comments.
Author Response
Dear reviewer,
We appreciate your involvement in reviewing our paper and your work, which helped us improve our article.
Sincerely yours,
Andreea Gherasim MD, PhD, lecturer
On behalf of the authors
Round 3
Reviewer 1 Report
Comments and Suggestions for Authors
There are too many errors around the structure of LL-37 and Figure 1. I recommend removing Figure 1 and all related discussion in the Section 2. Because all information presented here is wrong. Why the structure in Fig. 1 is tetrameric? The peptide is a monomer in the detergent micelle. And it is random coiled in aqueous solution and also monomer (Johansson J, Gudmundsson GH, Rottenberg ME, Berndt KD, Agerberth B. Conformation-dependent antibacterial activity of the naturally occurring human peptide LL-37. J Biol Chem. 1998 Feb 6;273(6):3718-24. doi: 10.1074/jbc.273.6.3718. PMID: 9452503). LL37 has no signal peptide. It is hCAP18 possess it. Taken into account the quality of the structural information, I do not believe in the other information presented in this review.
Author Response
Dear Reviewer,
Thank you for your detailed comments and time spent reviewing the manuscript.
Comment 1. There are too many errors around the structure of LL-37 and Figure 1. I recommend removing Figure 1 and all related discussion in the Section 2. Because all information presented here is wrong. Why the structure in Fig. 1 is tetrameric?
We reviewed the comments on the section on the structure of the LL-37 molecule. After review, we believe this section does not make an essential contribution to the overall context of the manuscript which addresses the relationships between the cathelicidin and the components of metabolic syndrome. Therefore, we decided to remove this section, to better focus on the functional and application aspects of the LL-37 in the components of the metabolic syndrome, which are more relevant to the main purpose of our paper.
Comment 2 The peptide is a monomer in the detergent micelle. And it is random coiled in aqueous solution and also monomer (Johansson J, Gudmundsson GH, Rottenberg ME, Berndt KD, Agerberth B. Conformation-dependent antibacterial activity of the naturally occurring human peptide LL-37. J Biol Chem. 1998 Feb 6;273(6):3718-24. doi: 10.1074/jbc.273.6.3718. PMID: 9452503).
Thank you for your input. We recognize the results from Johansson et al. (1998), which indicate that LL-37 is random-coiled and monomeric in aqueous solutions under particular conditions of low ionic strength and low concentration. Nonetheless, further research has demonstrated that LL-37 can oligomerize under varying conditions, including physiological ionic strength or slightly elevated concentrations.
- Aqueous Solutions: LL-37 mostly exists as a random coil and monomer in low ionic strength environments (Johansson et al.), but its structural flexibility facilitates oligomerization under physiological conditions, as revealed by :
- Tomasinsig L, Pizzirani C, Skerlavaj B, et al. The human cathelicidin LL-37 modulates the activities of the P2X7 receptor in a structure-dependent manner. J Biol Chem. 2008;283(45):30471–30481];
- Polcyn-Adamczak M, Niemir ZI. Cathelicidin – its structure, function, and the role in autoimmune diseases. Adv Cell Biol. 2014;4(2):83-96.;
- Bahar AA, Ren D. Antimicrobial peptides. Pharmaceuticals. 2013;6(12):1543-1575.
- Structural Flexibility: Verjans ET, Zels S, Luyten W, et al. Molecular mechanisms of LL-37-induced receptor activation: An overview. Peptides. 2016;85:16-26)
Comment 3. LL37 has no signal peptide. It is hCAP18 possess it. Taken into account the quality of the structural information, I do not believe in the other information presented in this review.
Our paragraph was as follows:
“Cathelicidin has three distinct domains: the signal peptide located at the N-terminal end, which plays a role in transporting the protein to its activation site where is processed by proteinase 3; the cathelin-like domain, well-conserved across species, which contributes to protein stabilization; and the antimicrobial domain located at the C-terminal end, responsible for disrupting bacterial membranes, neutralizing toxins, and participating in LL-37's functions within the immune system.”
We have not explicitly stated that LL-37 has three distinct domains, but our phrase could lead to misinterpretation as the word we used “cathelicidin” could be ambiguous in this context of its frequent use to describe the mature peptide LL-37..
We appreciate the valuable suggestions that helped us clarify the direction of the work and hope that the changes made will improve the quality and clarity of the manuscript.
Sincerely,
Andreea Gherasim
On behalf of the Authors